# ViRL-TSC: Enhancing Reinforcement Learning with Vision-Language Models for Context-Aware Traffic Signal Control

## Abstract

In real-world urban environments, traffic signal control (TSC) must maintain stability and efficiency under highly uncertain and dynamically changing traffic conditions. Although reinforcement learning (RL) has shown strong adaptability in dynamic environments, existing methods still depend on predefined state spaces and cannot directly perceive the environment. Consequently, they fail to exploit visual semantic information, which restricts their ability to generalize to unseen or evolving traffic conditions during training. To overcome these limitations, we introduce ViRL-TSC, a unified framework that integrates RL with Vision–Language Models (VLMs). A Foundation Model-Driven Visual Reasoning Engine (FM-VRE) fuses visual inputs with structured information matrices to generate high-level multimodal semantic representations of intersections. These representations are then processed by a Foundation Model-Driven Decision Evaluation Engine (FM-DEE), which integrates them with the RL agent's proposed actions. The RL policy ensures efficient control in scenarios encountered during training, while the VLM leverages logical reasoning and contextual analysis to handle rare events beyond the scope of RL training. By combining RL's task-specific policy optimization with the VLM's rich semantic understanding, ViRL-TSC maintains high efficiency during routine operations and selectively intervenes to enhance robustness under long-tail traffic conditions.

## 1 Introduction

Traffic congestion has become a critical challenge in modern urban governance, as it not only reduces travel efficiency and increases the risk of accidents but also contributes significantly to environmental pollution. Intelligent Traffic Signal Control (TSC) plays a central role in coordinating vehicle flows and improving transportation efficiency. However, conventional TSC methods often rely on predefined rules and fixed assumptions about traffic conditions, limiting their adaptability to complex and dynamic environments. In recent years, researchers have introduced deep reinforcement learning (RL) into traffic signal control, and a series of studies Noaeen et al. (2022); Zhao et al. (2024) have demonstrated clear advantages over traditional methods Lowrie (1990); Koonce & Rodegerdts (2008). The key strength of RL lies in its ability to learn policies for achieving long-term control objectives through direct interaction with dynamic environments.

Despite these advances, most existing TSC approaches face significant challenges in maintaining robustness and adaptability under highly dynamic and uncertain conditions of urban traffic Dulac-Arnold et al. (2021); Pang et al. (2024a); Chen et al. (2024). Current RL methods generally rely on static reward designs and observation states, which abstract away the semantically rich cues essential for context-aware decision-making. In real-world deployments, factors such as traffic control interventions, data loss due to RSU malfunctions, and rare but critical scenarios that are underrepresented during training pose substantial obstacles to the practical application of RL Noaeen et al. (2022). These issues collectively undermine the robustness and reliability of RL-based traffic signal control in real-world applications. Chen et al. (2023).

The emergence of large language models (LLMs) Chang et al. (2024) has opened new possibilities for introducing high-level reasoning and generalization capabilities into traffic signal control sys-

tems. Recent studies have explored integrating LLMs into TSC to enhance semantic reasoning and improve decision robustness Huang et al. (2022); Wang et al. (2024b); Pang et al. (2024b); Zhao et al. (2025a). By leveraging LLMs, agents can analyze intersection environments and make more contextually informed decisions. However, existing LLM-based approaches Huang et al. (2022); Wang et al. (2024b); Pang et al. (2024b); Zhao et al. (2025a) typically depend on templated or manually designed prompt descriptions of traffic scenes, which lack the richness and fidelity needed to accurately capture complex, real-world traffic situations. This limitation is particularly pronounced in visually grounded or ambiguous scenarios, often resulting in substantial information loss. Furthermore, as LLMs cannot natively process visual inputs, they fundamentally lack the ability to perform fine-grained visual semantic reasoning required for reliable TSC.

To address these limitations, a promising direction is to leverage Vision-Language Models (VLMs), which can directly interpret raw intersection images and construct high-level semantic representations, enabling end-to-end environmental understanding Da et al. (2023); Awais et al. (2025); Wang et al. (2025a). However, optimizing VLMs for task-specific control policies remains a significant challenge, as their large parameter size makes task-specific training on environment-specific data both computationally intensive and practically infeasible. This challenge highlights the complementary strength of RL in efficiently learning environment-adaptive policies. Motivated by this, we explore integrating the generalization and reasoning capabilities of foundation models with the efficient policy-learning ability of RL, enabling semantically grounded, causally aware environment modeling while generating robust and high-performance traffic signal control policies.

To this end, we propose **ViRL-TSC**, a unified framework that integrates RL with VLMs. RL agent provides efficient policy optimization, while VLMs enable multimodal scene understanding, allowing decision agent to refine RL actions based on environment visual information such as vehicle types, motion patterns, and so on. Specifically, a pre-trained RL agent generates actions from the information matrix. The **Foundation Model-Driven Visual Reasoning Engine**(FM-VRE) leverages multi-modal scene information to achieve a more comprehensive understanding and extract semantic representations. Subsequently, the **Foundation Model-Driven Decision Evaluation Engine** (FM-DEE) interprets these semantic representations to guide action selection. In scenarios encountered during RL training, the RL-generated actions are executed directly; in out-of-training scenarios, such as emergency vehicle (EMV) prioritization, the foundation model performs reasoning to produce appropriate actions, ensuring robust and context-aware control. By integrating RL and VLM, ViRL-TSC achieves efficient and reliable traffic signal control under real-world uncertainties. Our main contributions are as follows:

- We propose ViRL-TSC, a novel framework that integrates VLMs with RL to generate high-level semantic representations from intersection camera streams, surpassing traditional handcrafted traffic states and enhancing flexibility in understanding the environment. Combining RL and VLM allows the agent to make efficient decisions in scenarios encountered during training, while also enabling robust and context-aware traffic control in out-of-training scenarios.

- The FM-VREs leverages VLMs to process multi-modal data from both visual inputs and feature representations. Combined with the FM-DEE and chain-of-thought (CoT) prompting, ViRL-TSC effectively fuses these inputs to achieve more accurate, context-aware understanding of traffic scenes and generate appropriate decisions.

- Experimental results show that ViRL-TSC can reduce EMV waiting times by up to 88%, while the waiting time for other vehicles increases by no more than 1% when only a single emergency vehicle is passing. Additionally, further ablation studies on prompt design validate the effectiveness of the proposed prompting framework for traffic signal control.

## 2 PRELIMINARIES

As illustrated in Figure 1, we consider a standard four-way intersection to formalize key concepts in TSC. The network comprises four incoming and four outgoing roads, each with multiple lanes. A movement is defined as a traffic direction from an incoming lane to an outgoing lane, while a phase represents a set of non-conflicting movements operating simultaneously within a fixed interval; transitions between phases trigger a yellow light.

In our design, phases are grouped according to traffic flow directions, for example, from east to west, rather than lane-level movements, as current VLM capabilities in fine-grained lane detection are limited, making direction-based definitions more practical for real-world deployment.

To overcome the perception constraints of conventional traffic simulators, we develop a vision-enabled TSC simulator built upon SUMO Behrisch et al. (2011). In addition to the standard interface, the simulator integrates multi-view visual inputs, including a bird's-eye view for global traffic monitoring and directional roadside views emulating surveillance cameras commonly deployed at urban intersections. These multi-view inputs provide richer contextual information such as lane markings and emergency vehicle tracking, forming the basis for fine-grained scene understanding and enabling signal control strategies to incorporate both adaptability and safety awareness under diverse traffic conditions.

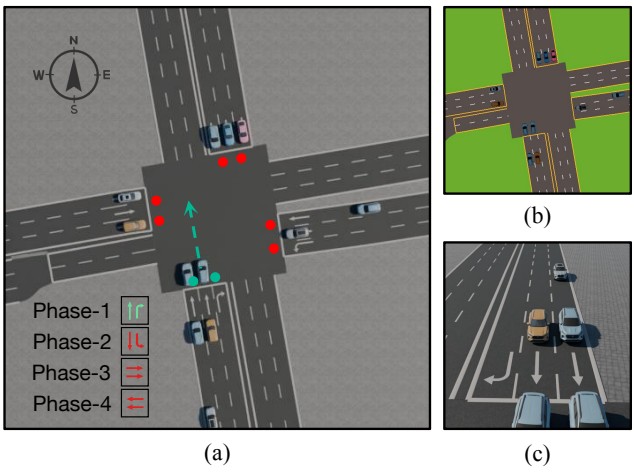

Figure 1: Simulation environment illustrating (a) the traffic signal phases and a bird's-eye view, (b) the SUMO simulation platform, and (c) rendered traffic probe data based on the intersection camera.

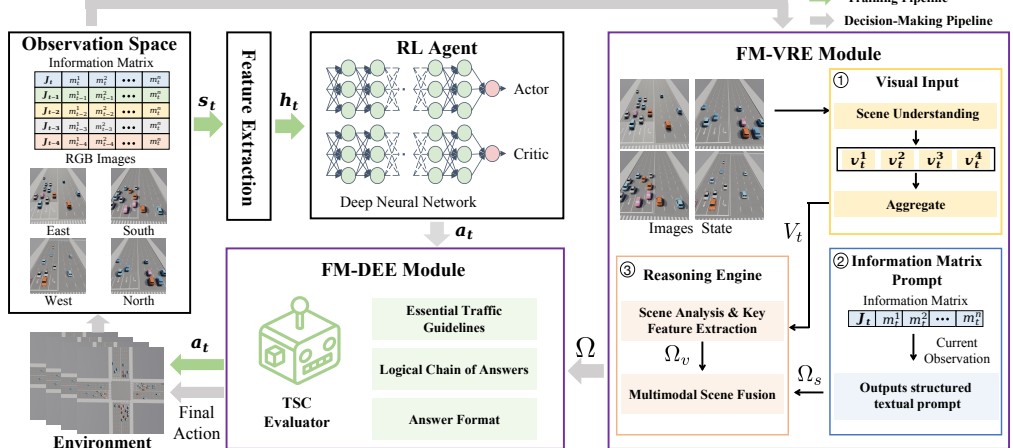

Figure 2: Overview of the ViRL-TSC framework. The RL Agent generates actions based on learned policies and is jointly trained within the closed-loop environment. The FM-VRE encodes multimodal inputs into semantic representations, and the FM-DEE integrates them to produce adaptive TSC decisions.

## 3 METHODOLOGY

We propose ViRL-TSC, a TSC framework that integrates RL with VLMs, as shown in Figure 2. The overall framework consists of three main components: an **RL Agent** that generates actions based on learned policies, a **FM-VRE** model responsible for processing multimodal data, and a **FM-DEE** that produces the final control strategy. Specifically, the RL Agent uses features extracted by the Feature Extraction module to generate actions according to its trained policy. The FM-VRE processes multimodal inputs, including multi-view intersection images and the Information

Matrix, and interprets them into natural language descriptions to obtain richer scene representations. The FM-DEE then integrates these semantic contexts with the RL-generated actions to assess their suitability: RL outputs are executed in routine control scenarios, while robust alternative actions are generated through logical reasoning in critical situations. This design enables ViRL-TSC to maintain real-time efficiency while ensuring robustness, safety, and interpretability under real-world uncertainties. A more detailed description of each component follows.

### 3.1 Problem Formulation

The TSC problem can be formulated as a Markov Decision Process (MDP) Chu et al. (2021), represented by the tuple $(\mathcal{S}, \mathcal{A}, \mathcal{P}, \mathcal{R}, \gamma)$. In this formulation, $\mathcal{S}$ denotes the state space, which captures both static and dynamic attributes of the intersection. $\mathcal{A}$ is the action space, representing signal phase selections. $\mathcal{P}$ defines the transition probability between states, $\mathcal{R}$ is the reward function, typically set as the negative average vehicle delay, and $\gamma \in (0, 1)$ is the discount factor.

At each time step $t$, the agent observes state $s_t \in \mathcal{S}$ and selects action $a_t \in \mathcal{A}$ based on a policy $\pi : \mathcal{S} \to \mathcal{A}$. The objective is to learn an optimal policy $\pi^*$ that maximizes the expected discounted cumulative reward:

$$\pi^*(s) = \arg\max_a \mathbb{E}\left[\sum_{t=0}^{\infty} \gamma^t r_t \mid s_0 = s, a_0 = a\right], \tag{1}$$

where $r_t = \mathcal{R}(s_t, a_t)$ denotes the immediate reward.

### 3.2 RL Agent Design

In the proposed method, the RL agent plays a pivotal role in making informed TSC decisions. This section outlines the three essential components that constitute the RL agent: state, action, and reward. Each component is critical for the agent's operation within the TSC environment, influencing its decision-making process in real-time traffic management.

**State:** At each timestep $t$, the intersection state is represented as:

$$\mathbf{J}_t = [\mathbf{m}_t^1, \ldots, \mathbf{m}_t^{12}] \in \mathbb{R}^{12 \times 7}, \tag{2}$$

where each movement $\mathbf{m}_t^i$ includes 7 features:

$$\mathbf{m}_t^i = \left[F_t^i, O_t^{\max,i}, O_t^{\text{mean},i}, I_i^{\text{type}}, L_i, I_t^{\text{green},i}, I_t^{\text{minG},i}\right] \tag{3}$$

with:

- $F_t^i$: average vehicle flow,
- $O_t^{\max,i}$: maximum occupancy ratio,
- $O_t^{\text{mean},i}$: mean occupancy ratio,
- $I_i^{\text{type}} \in \{0, 1, 2\}$: movement type (straight, left, right),
- $L_i$: number of lanes,
- $I_t^{\text{green},i} \in \{0, 1\}$: green light indicator,
- $I_t^{\text{minG},i} \in \{0, 1\}$: minimum green satisfied indicator.

Intersections with fewer than 12 movements are zero-padded. To capture temporal dynamics, the agent observes a 5-frame sequence, which spans approximately 25 s and effectively covers the typical time for vehicles to traverse the intersection:

$$s_t = [\mathbf{J}_{t-4}, \ldots, \mathbf{J}_t] \in \mathbb{R}^{5 \times 12 \times 7}. \tag{4}$$

**Action**: At the end of each time slot, the RL agent selects an action $a_t \in \boldsymbol{P}$, where each phase represents a directional traffic flow such as north–south or east–west. This abstraction shifts the control from lane-level granularity to direction-based signal phases, ensuring better compatibility with current vision-language models that are more effective at capturing aggregated directional flow patterns than detailed lane-specific states.

**Reward**: The RL agent is designed to improve traffic efficiency, and in this work, the average vehicle waiting time is used as the reward.

**RL Agent Training:** We adopt Proximal Policy Optimization (PPO) Schulman et al. (2017) due to its stability and effectiveness in dynamic traffic environments. To capture short-term temporal dependencies, a temporal encoder aggregates sequential traffic states $\mathbf{s}_t$ into embeddings $\boldsymbol{h}_t$, which are processed by a policy network $\pi_\theta$ to output signal-phase probabilities and a value network $v_\phi$ to estimate expected returns. This design enables the RL agent to learn adaptive, temporally-aware control policies. For detailed information, please refer to the Appendix A.1.

### 3.3 Foundation Model-Driven Visual Reasoning Engine

To enable the VLM to better interpret the multimodal information of an intersection, we introduce a **F**oundation **M**odel-Driven **V**isual **R**easoning **E**ngine, which processes visual inputs and the information matrix to generate high-level semantic representations, thereby achieving more precise and context-aware perception. As illustrated in Figure 2, the FM-VRE is composed of three key components:

**(1) VLM-Derived Visual Input:** Cameras at an intersection monitor traffic from different directions independently. To obtain a complete view of the intersection, the information from these cameras must be integrated. For example, at a four-way intersection, data from the north, south, east, and west directions need to be combined. Let $\mathcal{I}_t^d$ denote the raw image captured from direction $d \in \{1, \ldots, D\}$ at timestep $t$. Each $\mathcal{I}_t^d$ is processed by a VLM encoder $f_{\text{VLM}}(\cdot)$ to extract a semantic embedding $v_t^d$:

$$v_t^d = f_{\text{VLM}}(\mathcal{I}_t^d), \tag{5}$$

where $v_t^d$ encodes high-level traffic semantics such as congestion levels, queue density, and rare critical events (e.g., emergency vehicle presence or lane blockages) for the given direction.

The per-direction embeddings are then aggregated to form a comprehensive scene-level representation $V_t$:

$$V_t = \text{Aggregate}\left(v_t^1, \ldots, v_t^D\right), \tag{6}$$

where the Aggregate$(\cdot)$ denotes concatenation or learned fusion across directional features. The resulting $V_t$ provides a unified semantic summary of the entire intersection, grounded in raw visual context.

**(2) Feature-Driven Prompt Construction:** Visual inputs are insufficient for accurately capturing traffic states, such as vehicle speeds or lane-specific counts. To address this, we incorporate a structured traffic feature matrix $\mathbf{J}_t$ as a complementary information source. To ensure temporal consistency and inference stability while avoiding noise from outdated data, only the current snapshot of $\mathbf{J}_t$ is encoded, excluding historical states. The movement features in $\mathbf{J}_t = [\mathbf{m}_t^1, \ldots, \mathbf{m}_t^{12}]$ are converted into concise, structured natural language representations interpretable by the foundation model and mapped to phase-level information, denoted as $\Omega_s$. This provides the agent with more granular and compact semantic cues, reducing the ambiguity inherent in vision-only inputs and enabling a consistent and robust understanding of the current traffic state.

**(3) LLM-Driven Reasoning Engine:** This module is designed to integrate visual information with the structured feature matrix. First, the rich visual data is analyzed to extract relevant features, such as the presence of EMVs or traffic control measures, forming the visual semantic embedding $\Omega_v$. It is important to note that data derived solely from $\Omega_v$ may be imprecise due to the limitations of current VLMs, which can reliably detect only vehicles near the camera and cannot infer precise attributes such as vehicle speed. To address this limitation, $\Omega_v$ is fused with the structured embedding $\Omega_s$, which provides complementary and accurate traffic information. By jointly modeling visual

---

**Algorithm 1** ViRL-TSC inference with FM-DEE

---

**Require:** RL policy $\pi_\theta$, max VLM refinement attempts $K$
1: Initialize TSC environment
2: **for** each timestep $t$ **do**
3:     Observe structured state $s_t$ and camera images $\mathcal{I}_t^d$
4:     $a_t \leftarrow \pi_\theta(s_t)$                               $\triangleright$ initial action from RL agent
5:     **for** $i \leftarrow 1$ **to** $K$ **do**
6:         Derive visual semantic summary $\Omega_v$ (from $V_t$)
7:         Convert $\mathbf{S}_t$ to natural-language description $\Omega_s$
8:         Query VLM $P_{\text{VLM}}$ with prompt $\Omega \leftarrow \{\Omega_s, \Omega_v\}$ to get $\mathcal{O}_t$
9:         Parse candidate action $\hat{a}_t$ from $\mathcal{O}_t$ (e.g., regex)
10:       **if** parsing succeeds **and** $\hat{a}_t \in \mathcal{A}$ **then**
11:           $a_t \leftarrow \hat{a}_t$
12:           **break**                 $\triangleright$ accept VLM suggestion and stop refinement
13:       **else**
14:           $a_t \leftarrow \pi_\theta(s_t)$              $\triangleright$ fallback to RL action if invalid
15:       **end if**
16:     **end for**
17:     Apply final action $a_t$ to the simulator
18:     Observe next state $s_{t+1}$
19: **end for**

---

and structured inputs, the reasoning core generates a coherent, context-aware representation of the environment, offering reliable guidance for policy optimization and enhancing the system's robustness and adaptability in dynamic traffic scenarios, while ensuring the accuracy of both semantic and structured information.

### 3.4 ENHANCING RL DECISIONS WITH FOUNDATION MODEL-DRIVEN DECISION EVALUATION ENGINE

To enhance the reliability of TSC in diverse and uncertain scenarios, we propose a **F**oundation **M**odel-Driven **D**ecision **E**valuation **E**ngine that refines the initial control actions $a_t$ generated by the RL agent. The evaluator leverages the Structured Multimodal Scene Prompt $\Omega$ from the FM-VRE module, which integrates visual-semantic information with traffic-state representations. It operates on a prompt structure consisting of three components: **Essential Traffic Guidelines**, which encode rule-based constraints and safety principles for signal control; a **Logical Chain of Answers**, guiding the foundation model through step-by-step reasoning to ensure consistent, explainable, and risk-aware decision refinement; and an **Answer Format**, which enforces a fixed output structure, enabling candidate actions to be extracted via regular expressions. Specific implementation details can be found in Appendix A.2. The action space derived from these outputs is aligned with the RL agent's original action space, ensuring compatibility and seamless integration. Leveraging these inputs, the FM-DEE performs context-aware assessment of the RL agent's proposed action, provides structured reasoning explanations, and generates a grounded decision aligned with real-world traffic conditions.

Formally, the FM-DEE produces a candidate decision sequence $\mathcal{O}_t$ by maximizing

$$p(\mathcal{O}_t \mid \Omega) = \prod_{i=1}^{|\mathcal{O}_t|} p_{\text{VLM}}(o_i \mid \Omega, o_{<i}),$$

where $o_i$ denotes the $i$-th token. Candidate actions extracted via regular expressions are validated and retried up to $K$ times if invalid; otherwise, the original RL action $a_t$ is retained to ensure stable and continuous control. As shown in Algorithm 1, under normal circumstances, the system follows the RL agent's proposed action, while in rare or unforeseen situations not encountered during training, the FM-DEE leverages its understanding of the real-world scene and performs reasoning to generate the final, context-aware decision.

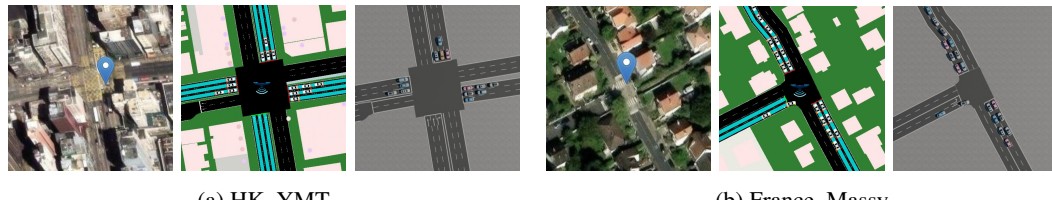

(a) HK, YMT                                      (b) France, Massy

Figure 3: Illustrations of two types of intersection environments: (a) a four-way intersection in the Hong Kong, YMT scenario, and (b) a T-shaped intersection in the France, Massy scenario.

## 4 EXPERIMENT

### 4.1 EXPERIMENTAL SETUP

All experiments are conducted in a custom TSC simulator that integrates SUMO with a multi-view camera rendering module. Each simulation episode follows standard urban traffic rules: green phases last at least 10 seconds, yellow phases 3 seconds, while red phases have no fixed upper bound. A minimum vehicle headway of 2.5 meters is enforced to ensure safety.

We conduct evaluations on two real-world intersections: Yau Ma Tei in Hong Kong and Massy in France, as shown in Figure 3. Yau Ma Tei represents a dense urban core with narrow lanes, restricted turns, and frequent congestion, whereas Massy is a suburban T-junction with wider lanes and simpler traffic flows, offering a complementary evaluation scenario.

For the VLM in ViRL-TSC, we adopt Qwen2.5 Bai et al. (2025) due to its open-source nature and efficient image encoding pipeline, which enables seamless integration with multi-view camera inputs. In offline mode, the inference time for a single decision step is within 3 seconds. The framework remains modular and can be extended with other foundation models.

Table 1: Comparison of model performance under two urban scenarios (Hong Kong Yau Ma Tei and France Massy). ↓ indicates lower is better.

| Category | Model | ATT ↓ | AWT ↓ | AETT ↓ | AEWT ↓ |
|---|---|---|---|---|---|
| **Hong Kong, Yau Ma Tei** | | | | | |
| Rule-based | FixTime | 67.63 ± 4.57 | 40.00 ± 2.28 | 82.67 ± 3.23 | 53.17 ± 2.14 |
| | Webster Webster (1958) | 56.26 ± 3.39 | 28.62 ± 1.23 | 59.67 ± 4.11 | 30.83 ± 1.79 |
| | MaxPressure Varaiya (2013) | 41.36 ± 2.22 | 13.33 ± 0.40 | 36.17 ± 2.39 | 8.83 ± 0.27 |
| RL-based | IntelliLight Wei et al. (2018) | 69.10 ± 2.86 | 13.73 ± 0.52 | 65.59 ± 2.80 | 8.14 ± 0.49 |
| | UniTSA Wang et al. (2024c) | **38.10 ± 1.28** [†] | **10.29 ± 0.50** [†] | 33.19 ± 1.92 | 5.17 ± 0.35 [§] |
| | CCDA Wang et al. (2024a) | 41.60 ± 2.02 | 11.23 ± 0.45 [‡] | 36.24 ± 1.45 | 5.65 ± 0.31 |
| | Vanilla-RL (Ours) | 39.54 ± 1.39 [‡] | 11.51 ± 0.16 [§] | 41.03 ± 0.55 | 13.22 ± 1.19 |
| VLM-based | Vanilla-VLM | 62.45 ± 8.76 | 17.62 ± 2.45 | 27.79 ± 3.79 [‡] | 5.86 ± 0.69 |
| | VLMLight Wang et al. (2025b) | 45.42 ± 7.18 | 16.68 ± 3.10 | 32.00 ± 2.86 [§] | 2.57 ± 0.93 [‡] |
| | **ViRL-TSC (Ours)** | 41.05 ± 0.73 [§] | 12.73 ± 0.75 | **25.67 ± 3.3** [†] | **1.50 ± 1.81** [†] |
| **France, Massy** | | | | | |
| Rule-based | FixTime | 75.84 ± 4.46 | 28.19 ± 1.58 | 73.60 ± 4.62 | 27.80 ± 1.06 |
| | Webster Webster (1958) | 68.92 ± 2.15 | 20.89 ± 0.79 | 65.20 ± 3.04 | 19.80 ± 0.61 |
| | MaxPressure Varaiya (2013) | 64.82 ± 4.01 | 15.25 ± 0.86 | 72.40 ± 2.83 | 22.40 ± 0.92 |
| RL-based | IntelliLight Wei et al. (2018) | 69.10 ± 2.86 | 13.73 ± 0.52 | 65.59 ± 2.80 | 8.14 ± 0.49 |
| | UniTSA Wang et al. (2024c) | **57.84 ± 1.91** [†] | 10.91 ± 0.54 [‡] | 64.90 ± 2.28 | 9.81 ± 0.41 |
| | CCDA Wang et al. (2024a) | 62.83 ± 2.42 | 11.19 ± 0.38 [§] | 58.80 ± 2.40 | 7.20 ± 0.49 |
| | Vanilla-RL (Ours) | 59.29 ± 1.41 [‡] | **10.77 ± 0.18** [†] | 58.08 ± 2.55 | 12.27 ± 0.46 |
| VLM-based | Vanilla-VLM | 68.40 ± 6.91 | 18.25 ± 2.01 | 53.01 ± 8.34 [§] | 3.52 ± 0.67 [§] |
| | VLMLight Wang et al. (2025b) | 62.18 ± 0.58 | 13.12 ± 0.13 | **47.32 ± 6.29** [†] | **2.04 ± 2.17** [†] |
| | **ViRL-TSC (Ours)** | 61.89 ± 0.94 [§] | 12.93 ± 0.37 | 49.20 ± 0.92 [‡] | 2.53 ± 0.50 [‡] |

*Ranking markers:* [†] best, [‡] second, [§] third.

Table 2: Performance under temporary traffic control at Hong Kong, Yau Ma Tei. ↓ indicates lower is better.

| Method | ATT ↓ | AWT ↓ | AETT ↓ | AEWT ↓ |
|---|---|---|---|---|
| FixTime | $82.60 \pm 0.04$ | $55.49 \pm 0.21$ | $79.67 \pm 4.67$ | $50.67 \pm 4.17$ |
| Vanilla-RL | $50.64 \pm 1.22$ | $20.51 \pm 0.87$ | $61.75 \pm 2.47$ | $29.59 \pm 2.00$ |
| VLMLight | $50.00 \pm 0.08$ | $20.23 \pm 0.11$ | $\mathbf{34.35 \pm 1.91}$ | $5.67 \pm 1.65$ |
| ViRL-TSC (Ours) | $\mathbf{45.58 \pm 1.56}$ | $\mathbf{16.67 \pm 0.48}$ | $35.59 \pm 1.29$ | $\mathbf{6.13 \pm 0.06}$ |

## 4.2 COMPARED METHODS

We evaluate our approach against a diverse set of baselines, including rule-based, RL-based, and VLM-based methods. Traditional approaches comprise FixTime, Webster Webster (1958), and MaxPressure Varaiya (2013). The RL-based baselines consist of IntelliLight Wei et al. (2018), UniTSA Wang et al. (2024c), and CCDA Wang et al. (2024a). We additionally include Vanilla-RL, which corresponds to the same RL backbone used in ViRL-TSC but without any VLM integration; this baseline isolates the contribution of VLM-guided reasoning. For VLM-based comparison, we adopt a Vanilla-VLM baseline that directly applies VLM outputs to decision-making without reasoning or iterative refinement. We further incorporate VLMLight Wang et al. (2025b), a vision-based TSC method that leverages VLMs for scene understanding while performing traffic signal control using standard RL mechanisms under typical scenarios. This provides a stronger comparison point to evaluate the benefit of our full VLM-guided RL framework.

## 4.3 EVALUATION METRICS

We assess system performance using four metrics: Average Travel Time (ATT), the mean time for all vehicles to reach their destinations; Average Waiting Time (AWT), the mean time vehicles remain nearly stationary (speed $< 0.1$ m/s); Average Emergency Travel Time (AETT), the mean travel time for emergency vehicles; and Average Emergency Waiting Time (AEWT), the mean waiting time for emergency vehicles. These metrics jointly reflect overall traffic efficiency and emergency vehicle prioritization.

## 4.4 PERFORMANCE COMPARISON

As shown in Table 1, we evaluate multiple TSC strategies across diverse urban environments, including the dense grid network in Yau Ma Tei, Hong Kong, and a T-shaped intersection in Massy, France.

**Routine Traffic Efficiency** Under normal traffic conditions, RL-based methods such as UniTSA, CCDA, and Vanilla-RL substantially outperform FixTime and rule-based strategies. For instance, in the Hong Kong scenario, Vanilla-RL achieves an AWT of $11.51$ s, roughly $60\%$ lower than Fix-Time's $40.00$ s. A similar pattern is observed in the France scenario, where Vanilla-RL reaches $10.77$ s AWT compared to Webster's $20.89$ s, demonstrating the effectiveness of adaptive control policies trained for specific scenarios. Compared with Vanilla-VLM, ViRL-TSC further reduces the average vehicle waiting time by approximately $28\%$ in Hong Kong Yau Ma Tei, and $29\%$ in France Massy, highlighting the benefit of leveraging RL decisions in routine traffic while enhancing overall performance. Compared with VLMLight, ViRL-TSC benefits from multi-source feature fusion rather than relying solely on visual inputs, leading to more accurate decisions and consistently lower delays across different traffic conditions. In practice, ViRL-TSC reduces AWT from $16.68$ s to $12.73$ s in Hong Kong, Yau Ma Tei and from $13.12$ s to $12.93$ s in France, Massy.

**Emergency Vehicle Efficiency** By incorporating visual semantics and multimodal reasoning, ViRL-TSC achieves both accurate traffic state understanding and robust EMV handling. In the Hong Kong experiment, ViRL-TSC reduces the AEWT to $1.50$ s, an improvement of approximately $88\%$ over Vanilla-RL at $13.22$ s, ensuring rapid EMV passage while maintaining network efficiency. In the France Massy intersection, ViRL-TSC lowers AEWT to $2.53$ s compared to Vanilla-RL at $12.27$ s,

Table 3: Ablation results on Hong Kong, Yau Ma Tei, impact of structured reasoning components and corresponding inference time.

| Modules | | | Metrics | | | |
|---|---|---|---|---|---|---|
| Answer Format | Logical Chain | Guidelines | AWT ↓ | AWT ↓ | AETT ↓ | AEWT ↓ |
| ✓ | ✗ | ✗ | 48.88 | 19.92 | 40.83 | 11.33 |
| ✓ | ✗ | ✓ | 44.90 | 16.12 | 33.67 | 4.67 |
| ✓ | ✓ | ✓ | **41.05** | **12.73** | **25.67** | **1.50** |

while keeping AWT within about 2 s of the RL baseline. This indicates that prioritizing EMV passage incurs only a minor cost to overall traffic delay. In addition, both VLMLight and ViRL-TSC leverage VLMs to identify EMVs, providing essential semantic information to support policy decision-making.

Overall, RL-based methods perform well under normal traffic but fail to adapt to EMVs. ViRL-TSC addresses this by combining RL decision-making with VLM reasoning, cutting EMV delays by over 80% while maintaining network stability.

## 4.5 PERFORMANCE UNDER TEMPORARY TRAFFIC CONTROL

To assess robustness under non-stationary conditions, we introduce a 50-second period of temporary traffic control on the Phase-1 approach at the Yau Ma Tei intersection. As shown in Table 2, FixTime and Vanilla-RL methods exhibit clear performance degradation under this disruption, especially in terms of AWT and AEWT. VLMLight benefits from visual perception but remains unstable due to its reliance on vision-only cues. In comparison, ViRL-TSC delivers the strongest overall performance. Its AWT is about 19% lower than that of VLMLight, and its ATT is reduced by roughly 9%. More importantly, ViRL-TSC continues to ensure fast emergency vehicle clearance even during traffic-control interventions, demonstrating that multi-source information fusion enables robust and reliable decision-making when normal traffic patterns are disrupted.

## 4.6 ABLATION ANALYSIS

To investigate the impact of each prompt component on the performance of the FM-DEE, we conducted ablation studies at the Yau Ma Tei intersection in Hong Kong. The prompt consists of three elements: (1) essential hints for decision-making (Guidelines), (2) Logical Chain of Answers (Logical Chain ), and (3) answer format. Since the VLM must understand the environment and respond in a specific format, the traffic scenario description and answer format are considered fundamental components. Therefore, our experiments focused on evaluating the effects of Guidelines and the logical chain. The results are summarized in Table 3.

The results indicate that relying solely on the answer format yields high AWT and AEWT, reflecting the limited reasoning capability of the VLM. Incorporating Guidelines substantially reduces waiting times, and further adding a logical chain enhances performance even more. When both Guidelines and logical reasoning are enabled, the system achieves optimal performance, with AWT and AEWT reduced to 41.05 s and 1.50 s, representing reductions of 36% and 87%. This demonstrates that structured semantic grounding, contextual Guidelines, and explicit logical reasoning complement each other, enabling the VLM to handle complex traffic scenarios effectively while improving both efficiency and decision reliability in the FM-DEE.

## 4.7 MODEL STABILITY ANALYSIS

To further evaluate the stability and effectiveness of the proposed ViRL-TSC framework, we examine its decision-making behavior under varying traffic conditions, with a focus on EMV prioritization. Specifically, we evaluate how the VLM's performance varies with different numbers of EMVs and the extent to which it modifies the RL agent's decisions.

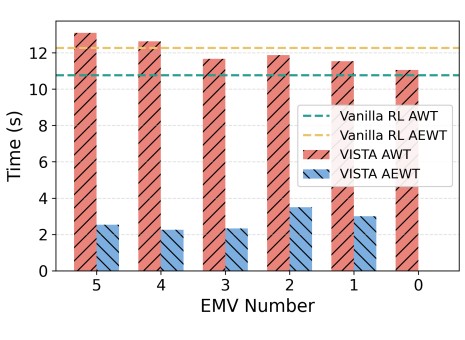
(a) Traffic Efficiency.

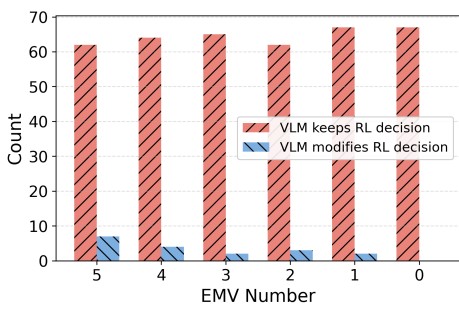
(b) Count of VLM preserved RL actions.

Figure 4: Impact of EMV number on ViRL-TSC performance.

As shown in Figure 4, prioritizing EMVs slightly increases AWT for regular vehicles, from $11.05\,\text{s}$ with no EMVs to $13.10\,\text{s}$ with five EMVs. Both the modification rate and AWT decrease as the number of EMVs declines, indicating adaptive adjustments and stable policy behavior.

Despite the marginal increase in AWT, EMVs are cleared rapidly, with expected waiting times maintained within 2–3 s, consistent with signal timing (full phase 5 s, yellow interval 3 s). The FM-DEE intervenes in only 3%–10% of cases, preserving the RL agent's decisions in most situations. This demonstrates ViRL-TSC's effectiveness: it relies on RL decisions in routine scenarios and intervenes selectively in critical situations to ensure rapid EMV passage.

## 5 CONCLUSION

This paper has proposed ViRL-TSC, a novel TSC framework that integrates VLM with RL. By exploiting the multimodal understanding and cross-domain generalization capabilities of VLMs, ViRL-TSC enhances the robustness and adaptability of RL-based TSC systems in complex, real-world environments. It optimizes RL-generated policies under dynamic and uncertain environments, enabling more context-aware and adaptive decision-making. Extensive experiments demonstrate that ViRL-TSC significantly narrows the performance gap between simulation-trained RL agents and real-world deployments, achieving stable and context-aware TSC. While visual perception still faces challenges when vehicles are distant from cameras, the information matrix can provide more precise complementary information.

Future research will focus on integrating visual and V2X data to enhance overall perception and decision-making capabilities. Additionally, large-scale deployment, multi-intersection coordination, and continual learning strategies will be explored to further improve system adaptability in open and dynamic traffic environments. More broadly, ViRL-TSC presents a task-agnostic paradigm that effectively combines foundation models with RL, opening new avenues for intelligent control applications in complex real-world systems.

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

# A   APPENDIX

## A.1   RL ALGORITHM

The training of the RL-based agent is a critical aspect of the proposed framework. In this study, the Proximal Policy Optimization (PPO) algorithm is employed to train the RL agent. PPO is chosen due to its suitability for policy-based reinforcement learning in environments with both discrete and continuous action spaces. This study utilizes two primary neural networks, namely a policy network denoted as $\pi_\theta$ and a value network denoted as $v_\phi$, where $\theta$ and $\phi$ are the respective network parameters. The policy network generates a probability distribution over the actions, given the current state, while the value network estimates the expected future return for that state. The traffic intersection observations $s_t$ are used as input for both networks, generating $\pi_\theta(s_t)$ and $v_\phi(s_t)$. Next, the following PPO objective is proposed by taking into account both a policy loss $\mathcal{L}_p(\theta)$ and a value function loss $\mathcal{L}_v(\phi)$:

$$F(\theta, \phi) = -\mathcal{L}_p(\theta) + \lambda \mathcal{L}_v(\phi), \tag{7}$$

where $\lambda$ is a hyperparameter that balances the two loss terms. Furthermore, the policy loss function $\mathcal{L}_p(\theta)$ is defined as:

$$\mathcal{L}_p(\theta) = \hat{\mathbb{E}}_\pi \left[ \min \left( J(\theta) \hat{A}_t, \text{clip} \left( J(\theta), 1 - \epsilon, 1 + \epsilon \right) \hat{A}_t \right) \right], \tag{8}$$

where $\hat{\mathbb{E}}_\pi$ denotes the empirical expectation under policy $\pi_\theta$ while $\epsilon$ represents the clipping range. Furthermore, $J(\theta)$ and $\hat{A}_t$ take the following form:

$$J(\theta) = \frac{\pi_\theta(a_t|s_t)}{\pi_{\tilde{\theta}}(a_t|s_t)}, \tag{9}$$

$$\hat{A}_t = r_{t+1} + \gamma v_\phi(s_{t+1}) - v_\phi(s_t), \tag{10}$$

where $\gamma$ is the discount factor that determines the present value of future rewards. $\pi_\theta(\cdot|\cdot)$ and $\pi_{\tilde{\theta}}(\cdot|\cdot)$ stand for the current and previous policy, respectively. The previous policy $\pi_{\tilde{\theta}}(\cdot|\cdot)$ is used as a reference to ensure that the policy updates are consistent with the previous policy.

Finally, the value function $\mathcal{L}_v(\phi)$ is computed as the mean-squared error between the predicted state values $v_\phi(s)$ and the actual discounted returns:

$$\mathcal{L}_v(\phi) = \hat{\mathbb{E}}_\pi \left[ \left( \hat{A}_t \right)^2 \right]. \tag{11}$$

By minimizing this loss, the agent learns to predict the expected return from each state more accurately, which in turn helps the agent to choose better actions.

## A.2   STRUCTURED PROMPT DESIGN FOR FM-DEE

To enable the LLM to reason about complex traffic control scenarios, we design a structured natural language prompt composed of the following four components, as shown in Figure 5:

1. **Role Definition:** The prompt begins by assigning the LLM the role of a TSC evaluator. This establishes the context and ensures the model understands its responsibilities in the decision-making process.

2. **Decision-Making Guidelines:** A concise list of operational rules and traffic principles is provided. These include prioritizing emergency vehicles, ensuring each intersection can select only one valid phase per time step, and following standard traffic safety practices. This guides the LLM toward rational and regulation-compliant actions.

3. **Reasoning Chain:** The prompt instructs the LLM to first evaluate the RL agent's proposed action based on the provided scenario context, explain whether it aligns with traffic guidelines, and, if needed, propose a revised action. This process encourages interpretable, rule-based reasoning rather than intuition-based responses.

**Role to be played: TSC Evaluator**

**Essential hints for decision-making:**
1. You must output a decision when you finish this task.
2. Your final output decision must be unique and not ambiguous.
3. You need to know your available actions and junction state before you make any decision.
4. Emergency vehicles have priority through intersections.
5. [...]

**Logical chain of answers:**
1. Decision:Traffic light decision-making judgment whether the Action is reasonable in the current state.
2. Explalanation: Your explanation about your decision, described your suggestions to the Crossing Guard. The analysis should be as detailed as possible, including the possible benefits of each action.
3.Final Action: An action is finally selected from the set of candidate phases.

**Answer Format:**
Format the output as JSON with the following keys:
```json
{
    "decision": string  // The final Acion ....
    "expalanations": string  // Your explaination about ...
}
```

Figure 5: The schematic diagram of LLM as a traffic evaluator, illustrates the three-stage architecture of LLM in traffic control: formulating thought programs, acquiring environmental information, and optimizing RL actions.

4. **Output Format Specification:** To support seamless integration into the TSC pipeline, the LLM is required to return a structured JSON response. This format ensures the output is machine-readable and can be directly parsed for downstream execution.

This structured prompt enables the LLM to bridge symbolic control logic and natural language reasoning, improving interpretability and robustness in traffic signal control decisions.

# B  RELATED WORKS

**RL-Based TSC Method.** TSC methods are generally categorized into traditional approaches and RL-based approaches. Traditional rule-based methods, including fixed-time control, Webster's method Koonce & Rodegerdts (2008), and SCATS Lowrie (1990), have been widely deployed in urban areas but often fail to adapt to the stochastic and dynamic nature of real-world traffic, resulting in suboptimal performance. To address this limitation, RL has been introduced into TSC to enable adaptive and real-time decision-making Oroojlooy et al. (2020); Pang et al. (2024a); Wang et al. (2024c). RL-based methods typically deploy trainable agents at intersections that dynamically adjust signals based on real-time traffic states and have demonstrated promising results in simulation studies. However, deploying these methods in real-world scenarios remains challenging due to issues such as long-tail events Su et al. (2022) and noisy or missing observations Feng et al. (2024), which are often underrepresented during training. To bridge the gap between controlled simulations and real-world complexity, this work proposes integrating foundation models into RL-based TSC frameworks, leveraging their generalization and representation capabilities to enhance robustness and adaptability.

**Foundation Models for TSC.** Developing adaptable TSC systems remains a significant challenge. Recent studies have primarily focused on integrating LLMs to improve generalization and adaptability Lai et al. (2023); Wang et al. (2024b); Hu et al. (2025). However, existing LLM-based approaches typically rely on handcrafted, scene-specific prompts and lack effective task-driven policy learning, limiting their flexibility and scalability in open-world environments Wu et al. (2024). Meanwhile, VLMs have demonstrated remarkable multimodal perception and reasoning capabilities in other domains Zhang et al. (2024); Xu et al. (2024), yet their application in TSC remains limited. To address these gaps, we investigate a combined framework of VLM and RL, aiming to enhance the robustness and adaptability of TSC systems through richer integration of visual and semantic information.

**VLM-Enhanced World models for Agent Control.** World models aim to simulate environment dynamics by predicting future states based on current observations and actions Ha & Schmidhuber (2018), enabling planning, reasoning, and imagination-based learning in RL. Traditional world models rely on explicit environment transition modeling Ha & Schmidhuber (2018); Hafner et al. (2023), whereas LLM have recently emerged as implicit world models due to their strong semantic reasoning and generalization capabilities Zhao et al. (2025a). However, in highly dynamic tasks such as TSC, existing VLMs mostly perform direct state prediction with simple input–output mappings, lacking the intermediate reasoning steps essential for complex decision-making. This limitation prevents them from addressing higher-level causal questions such as "What if a specific signal phase is applied?" or "How would the outcome change under a different past decision?". To overcome this, recent studies have introduced CoT reasoning to explicitly unfold intermediate inference steps and model the association–intervention–counterfactual causal chain, enabling multi-step and interpretable decision generation Zhang et al. (2024); Zhao et al. (2025b). Combining CoT with multimodal VLM perception from multi-directional cameras Awais et al. (2025) and precise structured traffic features, recent works explore integrating multimodal reasoning with reinforcement learning policies, significantly enhancing robustness and generalization under real-world challenges such as communication delays, observation noise, and long-tail events Pang et al. (2024b).

# C    EXPERIMENT DETAILS

## C.1    COMPARED METHODS

### C.1.1    RULE-BASED METHODS

We compare three classical rule-based traffic signal control strategies:

- **FixTime**: A fixed-time control policy where each traffic signal phase is assigned a constant duration. In our experiments, we adopt FixTime-30, where each phase is set to a fixed length of 30 seconds.

- **Webster**: A classical analytical method that determines the optimal signal cycle length and phase split based on traffic demand. It aims to minimize average vehicle delay through formula-based optimization.

- **MaxPressure**: A dynamic control algorithm that selects the phase with the maximum pressure, defined as the difference between upstream and downstream queues. This strategy seeks to improve overall traffic throughput and alleviate congestion.

### C.1.2    RL-BASED METHODS

We evaluate three representative reinforcement learning-based TSC methods:

- **IntelliLight**: A deep Q-learning-based method that selects the most appropriate signal phase every $5$ seconds. It incorporates a balanced experience replay buffer to address phase selection bias.

- **UniTSA**: An advanced RL model that employs junction matrices and state augmentation techniques to enhance generalization across heterogeneous intersections and traffic patterns.

- **CCDA**: A hybrid architecture combining a centralized critic with decentralized actors. It jointly adjusts all phase durations using discrete actions from $\{-6, -3, 0, 3, 6\}$ seconds, with decisions made every $10$ seconds to maintain smooth transitions.

- **Vanilla-RL**: A standard reinforcement learning baseline that directly learns a TSC policy from raw state representations without using any auxiliary modules, such as vision-language reasoning or expert priors. The agent interacts with the environment by observing the queue lengths and phase states, and optimizes its policy purely through reward feedback via trial-and-error exploration. This baseline reflects the performance of conventional RL methods under the same observation and reward settings, serving as a reference to highlight the effectiveness of incorporating high-level semantic cues in our proposed framework.

### C.1.3 VLM-Based Methods

We include two representative vision-language-based baselines that incorporate multimodal reasoning into traffic signal control:

- **VLMLight**: A recent VLM-enhanced TSC framework where an image-based VLM analyzes intersection scenes to provide semantic guidance for signal control. The system uses the VLM to interpret key visual cues—such as vehicle type, spatial density, and abnormal events—and switches between VLM-guided decisions and an RL policy. The RL agent operates on features extracted by the VLM, resulting in a tightly coupled perception–control design. While VLMLight demonstrates the potential of multimodal reasoning in TSC, its reliance on visual inputs alone limits its ability to integrate structured traffic features or multi-source data, and its dependence on cloud-based VLM inference makes it sensitive to communication delays.

- **Vanilla-VLM**: An ablation baseline where ViRL-TSC operates solely using the VLM component without any reinforcement learning module. Given an intersection image and structured state descriptions, the VLM produces a direct signal-phase decision through chain-of-thought reasoning. This baseline represents the standalone capability of foundation models in traffic signal control and allows us to evaluate how much improvement RL contributes in scenarios requiring long-term optimization or multi-step reasoning. The exact prompting structure used for Vanilla-VLM is provided in Appendix A.2 to ensure reproducibility.

### C.2 Reinforcement Learning Model Settings and Training Results

As shown in Table 4 summarizes the key hyperparameters used in training our reinforcement learning-based traffic signal control models. These include the trace-decay parameter $\gamma$, which determines the temporal discounting of future rewards; the policy update parameter $\lambda$; and the policy clipping range $\epsilon$ used to constrain policy updates during optimization. The length of state history $K$ indicates how many previous steps are incorporated into the current observation, enabling the model to capture temporal dynamics.

Table 4: List of RL Model Parameters Used in Simulation

| Variables | Value |
|---|---|
| Trace-decay Parameter $\gamma$ | 0.99 |
| Policy Update Parameter $\lambda$ | 1 |
| Policy Clipping Range $\epsilon$ | 0.2 |
| Length of State $K$ | 5 |

To evaluate training stability and performance convergence, we track the cumulative reward curves across training episodes. Figure 6 and Figure 7 show the training reward trajectories in two representative scenarios: France Massy and Hong Kong Yau Ma Tei (YMT), respectively. In both environments, the reward increases steadily as training progresses, indicating effective policy learning. The France Massy environment presents a moderate traffic pattern with consistent convergence, while the Hong Kong YMT environment exhibits more dynamic traffic conditions, leading to slightly higher reward variance. Nevertheless, both scenarios demonstrate satisfactory convergence performance.

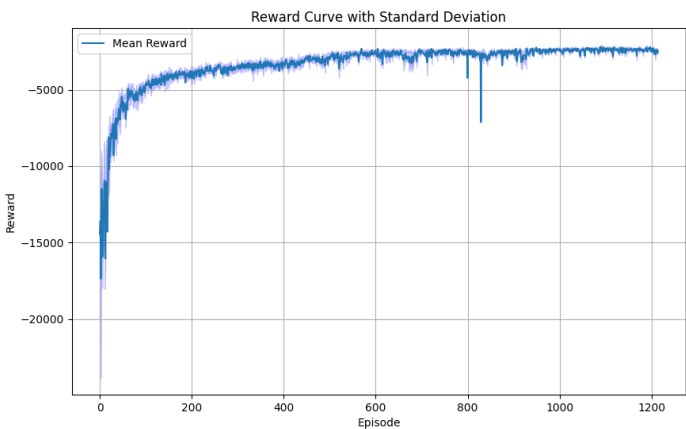

Figure 6: Training reward curve in France Massy.

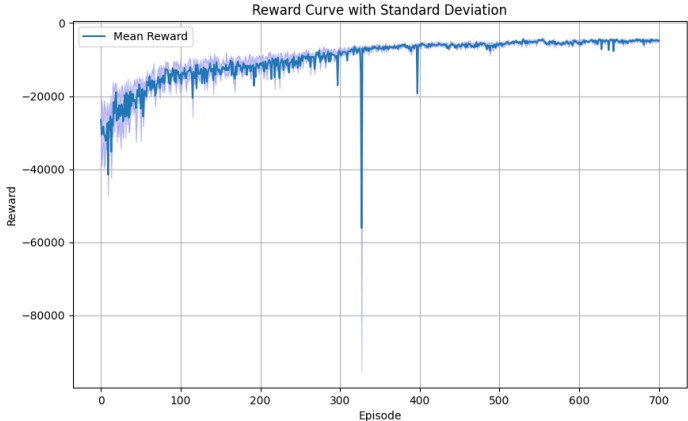

Figure 7: Training reward curve in Hong Kong YMT.

## C.3  CASE STUDY

Figure 8 illustrates a representative case study highlighting how ViRL's foundation model components collaboratively generate adaptive and context-aware signal decisions in response to emergency vehicle (EMV) events.

**FM-VRE Module.** This module integrates multi-modal traffic information using vision and language foundation models. First, the Vision-Language Model analyzes the real-time intersection image, identifying critical elements such as vehicle types, directions, and the presence of an EMV. Simultaneously, the Feature-Derived Prompt provides structured traffic state data—such as signal phases, vehicle queues, and waiting times—which complements the visual input. These inputs are jointly interpreted by the Large Language Model, which performs reasoning to associate semantic elements with the physical signal phases and spatial layout of the intersection.

**FM-DEE Module.** Based on the inferred intent and situation context provided by FM-VRE, the FM-DEE determines the most appropriate signal action. It ensures that the selected phase gives priority to the EMV while maintaining intersection safety and minimizing disruption to other traffic. By aligning multi-modal perception with structured control logic, this evaluator bridges human-like understanding with real-time signal operations.

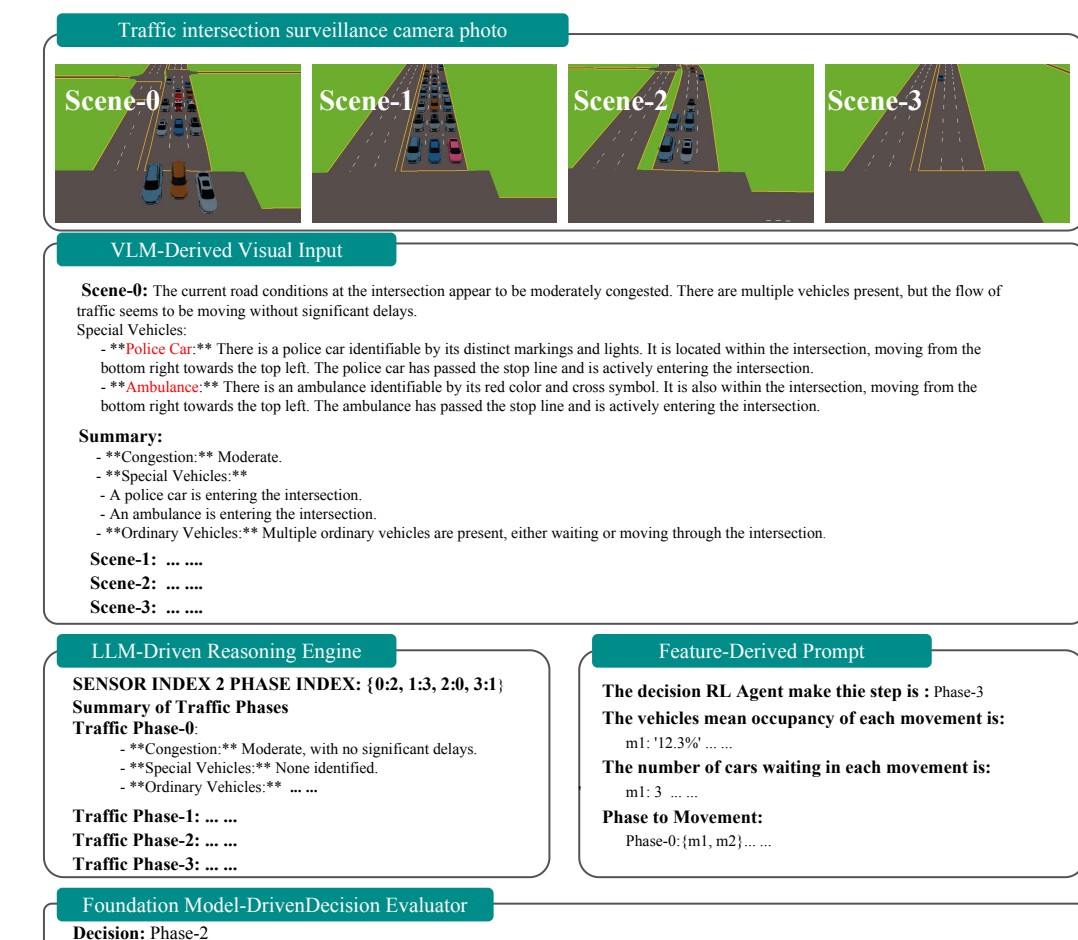

Figure 8: Case Study: Vi's Foundation Model-Based Decision Pipeline for Emergency Vehicle Prioritization.

This case demonstrates the interpretability, flexibility, and responsiveness of ViRL-TSC's foundation model-based pipeline, particularly under high-stakes scenarios such as emergency response.

## LLM USAGE

We used a large language model (GPT-5) solely for grammar checking and language refinement. The LLM did not contribute to research ideation, experimental design, implementation, or data analysis.

