# OpenReview forum: "ViRL-TSC: Enhancing Reinforcement Learning with Vision-Language Models for Context-Aware Traffic Signal Control"
_ICLR.cc/2026/Conference — Submitted to ICLR 2026_

### Official Review · Reviewer_anK7 · 2025-10-17

**Soundness:** 3
**Presentation:** 3
**Contribution:** 2
**Rating:** 4
**Confidence:** 5

**Summary:**

This paper proposed a unified framework that using RL with VLM called ViRL-TSC to robust traffic signal control. By combining RL's task-specific policy optimization with the VLM's rich semantic understanding, it maintains high efficiency during routine operations and selectively intervenes to enhance robustness under long-tail traffic conditions.

**Strengths:**

1. Proposed visual LLM integrates RL to solve traffic signal control problem. Which reduces the cost of perception deployment.

2. Proposed Foundation Model-Driven Decision Evaluation Engine (FM-DEE) that integrates visual input to percept special vehicle to increase robustness.

**Weaknesses:**

1. The experiments are only test in two single-intersection, the author should do the scalability experiment in large-scale intersections.

2. The idea is just using LLM and RL as a simple application in traffic signal control, it makes low novelty. The idea is similar to [1], but just change the perception module to visual. The idea is similar to [2], which is also using visual as perception, and i think the author just do some prompt engineer. The author should give some discuss about below two works and give some experiments compared to the two works.

[1] LLM-Assisted Light: Leveraging Large Language Model Capabilities for Human-Mimetic Traffic Signal Control in Complex Urban Environments. 2024 Arxiv

[2] VLMLight: Traffic Signal Control via Vision-Language Meta-Control and Dual-Branch Reasoning. 2025 NeuIPS

3. The author uses visual as input but use a structured traffic feature matrix J, i think the structured feature may percept from radar and the other non-visual sensors which is contradicts the full-visual solution mentioned by the author. The author should give more explanations.

4. There is no theoretical analysis.

5. The simulator is self-developed, not publicly available, and its results cannot be produced. The RL network, hyperparameters also not reported.

6. In real-world, the camera images we capture contain a lot of noise. For example, due to different weather conditions such as rain, snow and flog. Additionally, the captured images often have a lot of building information that is irrelevant to the lanes. Which is absolutely different from your self-developed simulator. Such information can cause the hallucination problem in your MLLM. The author should give more discussions or give a hallucination example.

**Questions:**

See weakness.

---

> ### Author Response · Authors · 2025-11-21
> **Authors' Response (1/2)**
>
> Thank you for your detailed and insightful comments. We truly appreciate the time you spent reviewing our work. Below, we provide point-by-point responses following the numbering of the identified Weaknesses and Questions. If you have any further questions or suggestions, we would be very happy to address them.
>
> **W1:**
> Thank you for the comment. Single-intersection settings are widely used in TSC research [1,2] as foundational benchmarks before scaling to coordinated networks, enabling controlled and interpretable evaluation. Our experiments already include two intersections with distinct geometries, demonstrating cross-layout generalization. In addition, ViRL-TSC follows a modular design—the RL agent and VLM refinement components can be directly incorporated into existing multi-intersection or multi-agent coordination frameworks, providing a clear path to scalability. Broader evaluations, including multi-intersection coordination, non-stationary demand, robustness, and transferability, will be explored in future work.
>
> [1] Wei, Hua, et al. “A survey on traffic signal control methods.” arXiv:1904.08117 (2019).
>
> [2] Zhao, Haiyan, et al. “A survey on deep reinforcement learning approaches for traffic signal control.” Engineering Applications of Artificial Intelligence 133 (2024).
>
> **W2:**
>  Thank you for the comment. ViRL-TSC is not an incremental variant of LLM-Assisted Light or VLMLight; the three approaches differ substantially in system architecture, information flow, and the role played by large models.
>
>  LLM-Assisted Light does not provide visual perception and relies solely on language inputs, meaning the agent cannot actively sense or interpret the traffic scene. VLMLight, in contrast, follows a VLM-first design: the VLM performs visual perception and scene interpretation, and the system decides whether the final action should come from the VLM or the RL agent. As a result, its decision-making remains primarily vision-driven. ViRL-TSC adopts a fundamentally different RL-first architecture. The RL agent always generates the initial action based on multi-source structured information, while the VLM is responsible for visual processing and provides semantic refinement only during inference. In practical deployments, these two designs lead to different operational properties.
>
> ViRL-TSC and VLMLight also differ in the types of information they can utilize. VLMLight relies mainly on visual embeddings, whereas ViRL-TSC integrates lane-level states, sensor readings, and controller metadata—structured information that current VLMs cannot reliably extract from images (e.g., the precise number of waiting vehicles).
>
> Finally, we have added VLMLight as a baseline and included an additional scenario experiment in the revised manuscript. Under temporary traffic-control conditions and using the same RL backbone, ViRL-TSC reduces average waiting time by more than 10%  compared with VLMLight. Overall, ViRL-TSC and VLMLight differ fundamentally in both architectural design and data-processing capabilities, rather than representing incremental variations of one another.
>
> **W3:**
> Thank you for the comment. ViRL-TSC is not intended to be a ``full-visual'' system. Current VLMs provide strong scene-level semantics but cannot reliably extract precise quantitative states (e.g., exact vehicle counts, lane occupancy, controller status), especially under occlusion or dense traffic. The structured feature matrix~$J$ supplies these accurate numerical signals to the RL agent, while visual input is used for high-level semantic reasoning. The two modalities are therefore complementary rather than contradictory.
>
> **W4:**
> Thank you for the comment. While this work provides empirical evidence for the benefits of VLM-RL integration, establishing a comprehensive theoretical foundation remains a significant and open challenge for the research community. We view this as a critical direction for future work, as the field of VLM/LLM-assisted methods in general currently operates with limited formal guarantees. Our focus is on system design and empirical validation, while safety is ensured by constraining the VLM to the predefined RL action set. We consider theoretical analysis an important direction for future work.
>
> **W5:**
> Thank you for the comment. The simulator is open-source, but since it is not the focus of this work, we did not emphasize it in the manuscript. The full implementation of ViRL-TSC will be released after the review process to ensure reproducibility and support the community.

---

> ### Author Response · Authors · 2025-11-21
> **Authors' Response (2/2)**
>
> **W6:**  Thank you for the comment. The detailed RL hyperparameters are reported in Appendix C.2 “Reinforcement Learning Model Settings and Training Results.” We also provide a clear description of the RL formulation and design choices in the main text, as this is a key component of our method. The underlying network architecture follows the standard PPO implementation from Stable Baselines3 without modification. For completeness and full reproducibility, we will release the full code and include the repository link after the review process.
>
> We agree that real-world camera images may contain noise (e.g., rain, fog, occlusion, background clutter), which can lead to unreliable visual perception and potentially induce hallucinations in VLMs. This limitation is precisely why ViRL-TSC is not designed as a purely visual system. In addition to the image input, the RL agent receives structured sensor-based features—such as lane-level vehicle counts and controller states—that current VLMs cannot robustly infer from images alone, especially under heavy congestion or limited visibility. These structured features provide accurate quantitative information, while the VLM contributes high-level semantic understanding; both modalities are therefore complementary.
>
> Our simulator already supports various environmental disturbances, including weather effects, noise, and visibility degradation. We plan to extend the experiments under these challenging conditions in future work.

---

### Official Review · Reviewer_BKxt · 2025-10-30

**Soundness:** 2
**Presentation:** 3
**Contribution:** 2
**Rating:** 4
**Confidence:** 5

**Summary:**

This paper proposes ViRL-TSC, a framework that integrates VLMs with RL to enhance context-aware traffic signal control in complex urban environments. The framework addresses the limitations of traditional and RL-based methods, which rely on predefined rules or static states and struggle with dynamic, real-world scenarios like emergency vehicle prioritization. ViRL-TSC uses a pre-trained RL agent for efficient policy optimization and a VLM-based reasoning engine to interpret multimodal data from cameras and traffic features, enabling high-level semantic understanding and robust decision-making. Experimental results demonstrate that the framework reduces emergency vehicle waiting times with minimal impact on regular traffic, thereby narrowing the performance gap between simulation-trained agents and real-world deployments.

**Strengths:**

1. **Integrated Multimodal Framework:** The paper integrate of VLMs with RL. This combination leverages RL for efficient policy optimization and VLMs for high-level, context-aware semantic understanding of complex traffic scenes from visual data.
2. **Enhanced Robustness in Rare Scenarios:** The framework improves robustness, particularly for unforeseen or rare events like emergency vehicle prioritization.
3. **Structured Reasoning for Decisions:** The incorporation of structured reasoning components, which guides the VLM to perform explicit, step-by-step reasoning. This ablation-studied approach substantially enhances decision reliability and performance over using a VLM without such guidance.

**Weaknesses:**

- **Limited Novelty**: The paper’s contribution appears incremental rather than groundbreaking. Prior work, such as [1], has already integrated VLMs with deep DRL model TSC, including specialized mechanisms for emergency vehicle response. The present study replicates most of this paradigm and only introduces a justified alignment mechanism between VLMs and DRL policies. Unfortunately, this alignment is also problematic (see detailed questions).
- **Limited baselines**: The experimental design omits several critical and directly comparable baselines. Recent studies have demonstrated the use of large language models (LLMs) [2–4] and VLMs [1] for adaptive and interpretable TSC, achieving strong generalization across diverse urban environments. However, these representative methods are not included for comparison. The omission prevents readers from evaluating whether the proposed method further improves interpretability and robustness.
- **Insufficient experiments**: The experimental evaluation lacks depth and scalability analysis. Most TSC studies assess their frameworks across multiple intersections (typically 12 or more) to capture complex traffic interactions and coordination challenges. In contrast, this paper’s experiments are confined to a single intersection, limiting the external validity of the findings. The absence of large-scale or heterogeneous network tests raises doubts about the method’s practicality in real-world deployments. Furthermore, key metrics such as transferability across layouts, robustness under non-stationary demand, and computational overhead are missing, further weakening the empirical evidence.

**Questions:**

- Existing studies [5, 6] have already established RL as the standard baseline for TSC agents. Therefore, Section 3.2, which primarily reintroduces the RL formulation, appears to be part of the background rather than the core methodological contribution.
- The paper describes a scene-level representation $V_t$ and its transformation into a visual semantic summary $\Omega_v$. Could the authors clarify this process? Specifically, is the transformation handled internally by the VLM, or is an additional language model or projection module implemented to interpret the visual embedding?
- The deep RL model outputs discrete actions, while the VLM operates through open-ended reasoning. How do the authors ensure that the alignment between the VLM’s reasoning process and the RL’s policy output is both semantically coherent and behaviorally consistent?
- The paper argues that RL models lack generalization and fail under unseen conditions. If so, what is the rationale for aligning the VLM’s decision-making with the RL’s policy output? Would this alignment not inherit RL’s generalization limitations rather than overcome them?

[1] Wang, Maonan, et al. "VLMLight: Traffic Signal Control via Vision-Language Meta-Control and Dual-Branch Reasoning." *arXiv preprint arXiv:2505.19486* (2025).

[2] Lai, Siqi, et al. "Llmlight: Large language models as traffic signal control agents." *Proceedings of the 31st ACM SIGKDD Conference on Knowledge Discovery and Data Mining V. 1*. 2025.
[3] Yuan, Zirui, Siqi Lai, and Hao Liu. "Collmlight: Cooperative large language model agents for network-wide traffic signal control." *arXiv preprint arXiv:2503.11739* (2025).

[4] Zou, Xingchen, et al. "Traffic-r1: Reinforced llms bring human-like reasoning to traffic signal control systems." *arXiv preprint arXiv:2508.02344* (2025).

[5] Wei, Hua, et al. "Colight: Learning network-level cooperation for traffic signal control." *Proceedings of the 28th ACM international conference on information and knowledge management*. 2019.

[6] Wei, Hua, et al. "Presslight: Learning max pressure control to coordinate traffic signals in arterial network." *Proceedings of the 25th ACM SIGKDD international conference on knowledge discovery & data mining*. 2019.

---

> ### Author Response · Authors · 2025-11-21
> **Authors' Response (1/2)**
>
> Thank you for your detailed and insightful comments. We truly appreciate the time you spent reviewing our work, and we have carefully addressed each point you raised. If you have any further questions, we would be happy to discuss them.
>
> **W1:**
> Thank you for the comment. ViRL-TSC is not an incremental variant of VLMLight [1]; it adopts a fundamentally different decision mechanism and system design.
>
> First, VLMLight adopts a VLM-first decision pipeline, where the VLM makes the primary decision and RL is used only as a fallback. In contrast, ViRL-TSC follows an RL-first design, with the VLM used solely for semantic refinement during inference. This ensures that the system remains fully functional even if cloud-based VLM access becomes unavailable, as the edge-side RL agent can operate independently.
>
> Second, the VLM component in ViRL-TSC supports multi-source information fusion, enabling the VLM agent to interpret structured inputs such as lane-level states, sensor readings, and traffic-control metadata. VLMLight, however, relies exclusively on visual inputs, and current VLMs still lack the capability to extract precise quantitative traffic information (e.g., the exact number of vehicles in each direction).
>
> Third, we have incorporated VLMLight as a baseline in the revised manuscript and added an additional scenario experiment. Under the same RL backbone, ViRL-TSC consistently outperforms VLMLight—achieving over 10\% improvement in temporary traffic-control scenarios—confirming that the performance gain results from fundamental architectural differences rather than incremental refinements.
>
> In summary, ViRL-TSC is an RL-first framework augmented with VLM-based refinement, offering stronger robustness, multi-source integration, and deployability, and is clearly distinct from the VLM-first, vision-only architecture of VLMLight.
>
> [1] VLMLight: Traffic Signal Control via Vision-Language Meta-Control and Dual-Branch Reasoning. 2025 NeuIPS.
>
> **W2:**
> Thank you for the helpful suggestion. In response, we have added  VLMLight [1], a recent 2025 VLM-based TSC method, as a strong and up-to-date baseline. To further strengthen the comparison, we also introduce an additional scenario involving a temporary traffic-control closure, providing a more comprehensive evaluation against this 2025 VLM-driven baseline.
>
> This scenario cannot be handled by the RL-only method (Vanilla-RL) because the closure is not encoded in the numerical state features. This highlights the necessity of visual semantic understanding in ViRL-TSC. We have included this experiment in the revised manuscript, and the quantitative results are shown below:
>
> | **Method**     | **ATT ↓**        | **AWT ↓**        | **AETT ↓**        | **AEWT ↓**        |
> |----------------|------------------|------------------|-------------------|-------------------|
> | FixTime        | 82.60 ± 0.04     | 55.49 ± 0.21     | 79.67 ± 4.67      | 50.67 ± 4.17      |
> | Vanilla-RL     | 50.64 ± 1.22     | 20.51 ± 0.87     | 61.75 ± 2.47      | 29.59 ± 2.00      |
> | VLMLight       | 50.00 ± 0.08     | 20.23 ± 0.11     | **34.35 ± 1.91**  | 5.67 ± 1.65       |
> | **ViRL-TSC**   | **45.58 ± 1.56** | **16.67 ± 0.48** | 35.59 ± 1.29      | **6.13 ± 0.06**   |
>
> As shown in the table, both rule-based (FixTime) and RL-only methods experience clear degradation under traffic-control disruptions. VLMLight benefits from visual perception but remains unstable due to its reliance on vision-only cues. In contrast, ViRL-TSC achieves the strongest overall performance. Its AWT is approximately **19% lower** than VLMLight, while still ensuring fast emergency-vehicle clearance. These results demonstrate that multi-source semantic fusion enables ViRL-TSC to maintain robustness when normal traffic patterns are disrupted.
>
> [1] VLMLight: Traffic Signal Control via Vision-Language Meta-Control and Dual-Branch Reasoning, NeurIPS 2025.
>
> **W3:**
> Thank you for the comment. Single-intersection setups are commonly used in TSC research [1,2] as baseline benchmarks before scaling to coordinated networks, allowing controlled and interpretable evaluation. Our experiments already include two intersections with distinct geometries, demonstrating cross-layout generalization. Moreover, ViRL-TSC is modular—the RL agent and VLM refinement components can be directly integrated into existing multi-intersection or multi-agent coordination frameworks, enabling straightforward scalability. Larger-scale studies, including multi-agent coordinated control and more complex traffic scenarios, will be further explored in future work.
>
> [1] Wei, Hua, et al. "A survey on traffic signal control methods." arXiv preprint arXiv:1904.08117 (2019).
>
> [2] Zhao, Haiyan, et al. "A survey on deep reinforcement learning approaches for traffic signal control." Engineering Applications of Artificial Intelligence 133 (2024): 108100.

---

> ### Author Response · Authors · 2025-11-21
> **Authors' Response (2/2)**
>
> **Q1:**  Section 3.2 is not intended as general background, but clarifies the specific RL formulation used in our framework. In ViRL-TSC, the action space is defined at the movement/approach level rather than using common phase-combination actions, which differs from traditional TSC RL designs. This section is therefore necessary to explain our RL design choices and how they integrate with the VLM refinement mechanism.
>
> **Q2:** The conversion from scene-level representation to a semantic summary is handled in two parts. The visual processing is performed internally by the VLM, which takes both the image and descriptive prompts as input. For the structured feature matrix, we provide explicit semantic descriptions (e.g., what each entry represents) to help the VLM agent correctly interpret non-visual information. This processing is carried out within the FM-VRE module.
>
> **Q3:** Thank you for the question. Semantic--behavioral alignment is strictly enforced in ViRL-TSC. The VLM never issues free-form text commands to control the traffic signals. As described in Algorithm~1, the VLM is constrained to select only from the predefined discrete action set $\mathcal{A}$, which corresponds exactly to the RL agent’s action space.
>
> During inference, the VLM receives (i) a structured description of the current scene and (ii) the full list of candidate actions with their semantic meanings, and is explicitly instructed to output only the ID of one valid action in $\mathcal{A}$. We parse the VLM output and, if the format is invalid, re-prompt up to $K$ times. If no valid action is produced, the system automatically falls back to the original RL action.
>
> As a result, the final executed action is always within the legal RL action space. The VLM can only refine RL-selected actions at a semantic level and cannot generate out-of-range behaviors, ensuring strict semantic--behavioral alignment and preventing control-level hallucinations.
>
> **Q4:**  When we mention that RL may generalize poorly, we refer to its instability when encountering rare or long-tail scenarios that were not covered during training, while RL remains highly effective under normal traffic conditions. In ViRL-TSC, RL serves as the primary controller, while the VLM provides semantic correction only in situations underrepresented during RL training (e.g., EMVs or local anomalies). The VLM selects alternative actions strictly from the same action set $\mathcal{A}$, correcting potential RL failures without expanding the action space. Therefore, aligning VLM reasoning with RL does not inherit RL limitations; instead, it strengthens robustness by combining RL efficiency with VLM semantic awareness.

---

### Official Review · Reviewer_Za98 · 2025-11-04

**Soundness:** 2
**Presentation:** 2
**Contribution:** 3
**Rating:** 4
**Confidence:** 3

**Summary:**

This paper investigates the integration of vision–language models (VLMs) with reinforcement learning (RL) for traffic signal control, aiming to improve decision-making under dynamic and uncertain urban conditions. The authors combine visual scene understanding with traffic state information to enable more context-aware and robust signal control in real traffic scenarios. The main contributions include: (1) a unified framework that combines reinforcement learning with vision–language models to improve context-aware traffic signal control; (2) a clear reasoning pipeline that integrates visual scene information with structured traffic features to refine decision-making; and (3) experimental results showing large improvements in emergency vehicle priority with only a small impact on normal traffic. The work demonstrates the potential of using multimodal reasoning to enhance the robustness of RL-based traffic control systems.

**Strengths:**

The main contributions include: (1) a unified framework that combines reinforcement learning with vision–language models to improve context-aware traffic signal control; (2) a clear reasoning pipeline that integrates visual scene information with structured traffic features to refine decision-making; and (3) experimental results showing large improvements in emergency vehicle priority with only a small impact on normal traffic.

**Weaknesses:**

However, there are several areas that require attention and improvement:
(1) The Introduction section does not clearly highlight the key gap in current TSC methods. The authors should explicitly explain why existing RL and LLM-based approaches cannot handle visual semantic information, and provide concrete examples to strengthen the motivation.

(2) The Compared Methods section does not clearly explain the Vanilla-VLM baseline. The authors should provide a clearer description of how this baseline is built, including the model setup and the prompt used, since the current statement that it directly uses VLM outputs lacks sufficient detail to understand or reproduce.

(3) In Fig. 2, the framework diagram does not clearly illustrate how the VLM-guided refinement contributes to policy improvement. It is unclear whether the refined actions are fed back into the RL learning process or only applied at inference time. The authors should clarify the closed-loop mechanism and show how the VLM intervention leads to measurable policy improvement, ideally through an explicit feedback or training signal in the diagram.

(4) The Experiments section lacks recent and strong VLM or LLM-based TSC baselines. Adding representative methods from 2023–2025 would make the comparison more fair and clearly demonstrate the advantages of the proposed approach.

(5) The Methodology section does not sufficiently justify the need to separate FM-VRE and FM-DEE into two modules. The authors should explain why a single-stage or end-to-end design is not adequate, and an ablation study is recommended to validate the architectural choice.

**Questions:**

(1) The Introduction section does not clearly highlight the key gap in current TSC methods. The authors should explicitly explain why existing RL and LLM-based approaches cannot handle visual semantic information, and provide concrete examples to strengthen the motivation.

(2) The Compared Methods section does not clearly explain the Vanilla-VLM baseline. The authors should provide a clearer description of how this baseline is built, including the model setup and the prompt used, since the current statement that it directly uses VLM outputs lacks sufficient detail to understand or reproduce.

(3) In Fig. 2, the framework diagram does not clearly illustrate how the VLM-guided refinement contributes to policy improvement. It is unclear whether the refined actions are fed back into the RL learning process or only applied at inference time. The authors should clarify the closed-loop mechanism and show how the VLM intervention leads to measurable policy improvement, ideally through an explicit feedback or training signal in the diagram.

(4) The Experiments section lacks recent and strong VLM or LLM-based TSC baselines. Adding representative methods from 2023–2025 would make the comparison more fair and clearly demonstrate the advantages of the proposed approach.

(5) The Methodology section does not sufficiently justify the need to separate FM-VRE and FM-DEE into two modules. The authors should explain why a single-stage or end-to-end design is not adequate, and an ablation study is recommended to validate the architectural choice.

---

> ### Author Response · Authors · 2025-11-21
> **Authors' Response (1/2)**
>
> We sincerely thank you for the detailed and constructive feedback. Your comments have greatly helped us improve the clarity and completeness of the manuscript. Below, we provide point-by-point responses to all of your concerns.
>
> **W1:**
> We have updated the Introduction to explicitly address the key limitations of existing TSC methods. Current RL-based approaches remain fragile under long-tail events and noisy observations because mainstream simulators provide only low-dimensional numerical features, meaning that RL agents never encounter visual semantic information during training. As a result, unless such cases are explicitly incorporated during training, RL models cannot recognize or respond to critical semantic events such as emergency vehicles, lane blockages, or temporary traffic control during deployment.
>
> Similarly, existing LLM-based methods (see Appendix B: Foundation Models for TSC) are unable to directly interpret visual scenes. They rely on manually crafted, scene-specific textual descriptions, which fail to capture the complexity and ambiguity of real-world traffic environments, limiting their scalability and robustness in open-world settings. Although VLMs offer stronger visual understanding, current VLMs still struggle to integrate multi-source traffic features and cannot independently perform the policy-level reasoning required for real-time traffic control.
>
> Motivated by these limitations, we propose a new framework that combines VLM-based visual semantic understanding with the efficient policy-learning capabilities of RL, enabling more robust and context-aware traffic signal control.
>
> **W2:**
> The Vanilla-VLM baseline corresponds to the setting in which ViRL operates solely with the VLM component, without the RL module. Likewise, the Vanilla-RL baseline isolates the RL module and does not involve any VLM information. We have clarified these configurations in the revised manuscript (see Appendix C.1.3, VLM-Based Methods). The full composition of the prompts used for Vanilla-VLM is also provided in Appendix A.2. In addition, we describe the complete inference procedure to ensure full reproducibility. We will also release all implementation code to further support reproducibility and facilitate future research.
>
> **W3:**
> The previous version of Fig.2 mainly illustrated the decision-making workflow. We have updated Fig.2 to clearly indicate that the RL agent must be trained, and to show that the VLM modules (FM-VRE and FM-DEE) are used only during the inference stage while interacting with the RL agent to support decision-making. In addition, the closed-loop mechanism is further clarified in Algorithm 1, which explains how the semantic refinements generated by the VLM are incorporated into the RL state representation.
>
> **W4:**
> Thank you for the helpful suggestion. In response, we have added VLMLight [1], a recent 2025 VLM-based TSC method, as a strong and up-to-date baseline. We have also included an additional scenario experiment to provide a more comprehensive comparison against this 2025 VLM-based baseline.
>
> [1] VLMLight: Traffic Signal Control via Vision-Language Meta-Control and Dual-Branch Reasoning. 2025 NeuIPS.
>
> **W5:**
> Thank you for the suggestion. The separation of FM-VRE and FM-DEE is intentional, as the two modules serve fundamentally different roles within the system. FM-VRE focuses on visual semantic extraction and visual–state alignment, which necessarily requires a VLM with strong visual capabilities. In contrast, FM-DEE is designed for policy reasoning and decision enhancement, a process that does not rely on visual input and can therefore leverage an LLM independently. Although VLMs possess some reasoning ability, merging visual processing and decision reasoning into a single module would reduce modularity and limit flexibility in replacing or improving individual components. For these reasons, we adopt a decoupled design that allows visual extraction and decision reasoning to operate independently, which forms an essential design principle of our framework.

---

> ### Author Response · Authors · 2025-11-21
> **Authors' Response (2/2)**
>
> **Q1:**
> Existing RL- and LLM-based TSC methods cannot process visual semantic information because they rely solely on numerical traffic states and lack any visual perception capability. LLM-based approaches operate only on text and therefore cannot interpret images. Using RL directly for vision-based TSC has not yet been explored, although related ideas have emerged in other domains, such as world-model-based control [1]. Furthermore, even with visual inputs, RL policies often struggle to generalize to unseen scenarios, such as unexpected road closures.
>
> [1] Hafner, D., Pasukonis, J., Ba, J. et al. Mastering diverse control tasks through world models. Nature 640, 647–653 (2025). https://doi.org/10.1038/s41586-025-08744-2
>
> **Q2:**
> Please refer to W2
>
> **Q3:**
> Please refer to W2.
> In short, VLM refinement is applied only during inference, while the RL agent is trained independently, ensuring a closed-loop interaction at inference time.
>
> **Q4:**
> Please refer to W4.
>
> **Q5:**
> Please refer to W5.

---

### Official Review · Reviewer_6Xah · 2025-11-11

**Soundness:** 3
**Presentation:** 3
**Contribution:** 2
**Rating:** 4
**Confidence:** 3

**Summary:**

This paper proposes to integrate RL with VLMs to improve the robustness and generalization for traffic signal control (TSC) tasks. Traditional RL-based TSC methods heavily rely on fixed state representations and struggle with unseen traffic scenarios, and the authors leverage the efficiency of a pretrained RL agent for routine traffic control, while the VLM selectively intervenes in long-tail, unseen situations. They conduct extensive experiments on two real-world intersections and demonstrate that their method can reduce the emergency vehicle waiting times.

**Strengths:**

1. Integrating VLMs with a pretrained RL agent for TSC is meaningful, as it combines the efficiency of RL-based methods with the generalization ability of foundation models.

2. The authors present extensive experimental results on two real-world intersections, comparing their approach with multiple baselines including rule-based, RL-based, and VLM-based methods.

3. The paper is clearly written, well organized, and easy to read.

**Weaknesses:**

1. While I do appreciate the idea of using foundation models to improve generalization in TSC, I am unclear whether the authors only consider emergency vehicle prioritization as the long-tail or unseen scenario. In real-world settings, short-term high traffic flow (e.g., due to local events) or traffic patterns that deviate from the regular 24-hour distribution may better represent long-tail situations. Could the authors evaluate their method under such scenarios?

2. The experiments are conducted on only two intersections, which may limit the significance and generality of the results. I understand that collecting real-world visual data from many intersections is difficult. However, since the authors have extended the SUMO simulator to a vision-enabled TSC simulator, they could evaluate their method on larger networks. For example, [1] provides datasets from Jinan (12 intersections) and Hangzhou (16 intersections). Moreover, some RL-based baselines such as CoLight [2] rely on communication and coordination among neighboring intersections, which could be relevant for comparison.

3. The decision cost is relatively high, which may hinder real-world deployment. According to line 344, a single decision step requires about three seconds for inference, and this cost could increase further if multiple intersections are controlled in parallel.


[1] A Survey on Traffic Signal Control Methods. https://traffic-signal-control.github.io/

[2] CoLight: Learning Network-level Cooperation for Traffic Signal Control.

**Questions:**

1. Which version of the Qwen model is used as the VLM backbone?

2. I believe Average Travel Time (ATT) and Average Waiting Time (AWT) are the two most important metrics for TSC. In Table 1, the **Vanilla-RL** (w/o VLM) seems to perform better than **ViRL-TSC**(w/ VLM) in ATT and AWT. Did I misunderstand your experimental setup? I hope the authors can clarify this.

3. Since the paper focuses on improving the generalizability of RL-based TSC, it might also be valuable to consider transferability, as smaller cities often lack sufficient data to train a good RL model. I do not expect the authors to include this in the current version, but I would like to discuss their problem formulation and experimental settings during the rebuttal phase to better understand the design choices and adjust my rating accordingly.

---

> ### Author Response · Authors · 2025-11-21
> **Authors' Response (1/2)**
>
> We sincerely thank you for your thorough and constructive comments. We have carefully considered each point and revised the manuscript accordingly. Below, we provide point-by-point responses following the numbering of the identified Weaknesses and Questions. We hope that our clarifications adequately address your concerns, and we are happy to provide further details if needed.
>
> **W1:**
> Thank you for the insightful comment. We fully agree that real-world long-tail scenarios extend beyond emergency-vehicle prioritization. Situations such as temporary traffic surges due to event dispersal can indeed be modeled through RL training, and RL-based approaches are generally capable of adapting to such demand fluctuations.
>
> However, some long-tail events stem from external interventions that cannot be observed or inferred from numerical traffic states by an RL agent. To illustrate this type of scenario, which requires visual–semantic reasoning rather than RL training alone, we introduce a 50-second temporary traffic-control closure on the Phase-1 approach at the Yau Ma Tei intersection.
>
> This scenario cannot be handled by RL-only methods because the closure is not encoded in the state features. It therefore highlights the necessity of visual understanding in ViRL-TSC. We have added this experiment to the revised manuscript. The quantitative results are shown below:
>
> | **Method**     | **ATT ↓**        | **AWT ↓**        | **AETT ↓**        | **AEWT ↓**        |
> |----------------|------------------|------------------|-------------------|-------------------|
> | FixTime        | 82.60 ± 0.04     | 55.49 ± 0.21     | 79.67 ± 4.67      | 50.67 ± 4.17      |
> | Vanilla-RL     | 50.64 ± 1.22     | 20.51 ± 0.87     | 61.75 ± 2.47      | 29.59 ± 2.00      |
> | VLMLight       | 50.00 ± 0.08     | 20.23 ± 0.11     | **34.35 ± 1.91**  | 5.67 ± 1.65       |
> | **ViRL-TSC**   | **45.58 ± 1.56** | **16.67 ± 0.48** | 35.59 ± 1.29      | **6.13 ± 0.06**   |
>
> As shown above, rule-based (FixTime) and RL-only methods suffer clear degradation under traffic-control disruptions. VLMLight benefits from visual cues but remains unstable due to its reliance on vision-only information. ViRL-TSC achieves the best overall performance, reducing AWT by approximately 19%  compared with VLMLight, while also maintaining fast emergency vehicle clearance. These results demonstrate that multi-source semantic fusion enables ViRL-TSC to remain robust when normal traffic patterns are disrupted.
>
> **W2:**
> We fully agree that multi-intersection coordination is a key challenge in real-world TSC. We focus on single-intersection scenarios for the following reasons. First, as highlighted in prior studies [1,2], single-junction control remains the fundamental building block of TSC research and an essential benchmark before scaling to larger coordinated networks, enabling a clearer understanding of model behavior. Second, to ensure generalizability, ViRL-TSC is intentionally designed as a modular framework in which the VLM controller and the RL controller can be seamlessly integrated into any multi-intersection coordination algorithm. Finally, our experiments already include two distinct intersection geometries, demonstrating that the framework adapts well to different layouts and possesses strong generalization potential.
>
> [1] Wei, Hua, et al. "A survey on traffic signal control methods." arXiv preprint arXiv:1904.08117 (2019).
>
> [2] Zhao, Haiyan, et al. "A survey on deep reinforcement learning approaches for traffic signal control." Engineering Applications of Artificial Intelligence 133 (2024): 108100.
>
> **W3:**
> Nevertheless, we would like to clarify that the proposed framework is designed under a cloud–edge collaborative architecture: the LLM is deployed in the cloud, while the RL agent operates on edge devices. Under this paradigm, simultaneous processing at multiple intersections is feasible, and the per-step decision latency remains stable, showing no significant increase compared with running a single intersection. If the hardware permits, the entire pipeline can also be deployed on edge devices. Recent studies have further demonstrated that modern LLMs can already run efficiently on edge hardware [3], which mitigates concerns regarding decision latency in large-scale deployments.
>
> [3] Yao, Yuan, et al. “Efficient GPT-4V level multimodal large language model for deployment on edge devices.” Nature Communications 16.1 (2025): 5509.

---

> ### Author Response · Authors · 2025-11-21
> **Authors' Response (2/2)**
>
> **Q1:**
> We use Qwen2.5-VL-Max as the backbone of the FM-VRE module and Qwen2.5-Max for the FM-DEE module in our framework. Both models belong to the Qwen2.5 series [4] and are accessed via the online API. We have also updated the manuscript accordingly to clarify these details.
>
> [4] Bai S, Chen K, Liu X, et al. Qwen2.5-vl technical report[J]. arXiv preprint arXiv:2502.13923, 2025.
>
> **Q2:**
> Thank you for the insightful observation. The slight decrease in ATT and AWT is expected due to the inherent trade-off in traffic control, where prioritizing one movement inevitably delays others [5]. In our framework, the VLM is intentionally instructed to prioritize emergency vehicles (EMVs) to validate its visual semantic understanding. When green time is allocated to the EMV approach, other directions naturally experience some delay—this is consistent with the predefined control objective rather than a model deficiency. Importantly, this objective can be modified directly through the Essential Traffic Guidelines in the FM-DEE module without any retraining.
>
> We have also added a new case study involving temporary traffic control. In such scenarios, the RL agent—lacking awareness of external interventions—may output invalid or inefficient actions. In contrast, ViRL-TSC uses visual-scene understanding to correctly recognize the traffic-control event and maintain efficient operation. This demonstrates that ViRL-TSC provides improved robustness in long-tail and externally influenced scenarios.
>
> [5] I. Alvarez et al., “Urban traffic control problem: a game theory approach,” CDC, 2008.
>
> **Q3:**
>  Our framework is designed with the RL and LLM components fully decoupled, so they do not require joint training. This allows the RL agent to be independently optimized for better generalization. Prior studies [6] have already explored training RL policies that can operate across multiple intersections, and such transferable RL models can be directly plugged into our framework. When doing so, the system can naturally adapt to different intersection layouts.
>
> The LLM-based modules also generalize well: the FM-VRE and FM-DEE modules only need updated environment descriptions to align with a new intersection, without any retraining of the LLM. This is demonstrated in our experiments, where the same LLM components are used for both a three-approach T-junction and a four-approach cross intersection, and only the RL agent is retrained for the new geometry.
>
> [6] Wang M, Xiong X, Kan Y, et al. UniTSA: A universal reinforcement learning framework for V2X traffic signal control[J]. IEEE Transactions on Vehicular Technology, 2024, 73(10): 14354-14369.

---

### Author Response · Authors · 2025-11-26
**General Response to Reviewers**

We thank all reviewers for their thoughtful evaluations. The main concerns raised are as follows:

1. Reviewers **anK7** and **BKxt** questioned the **novelty**, especially in comparison with **VLMLight (NeurIPS 2025)** [1] . We clarify that, relative to VLMLight, our work introduces two key innovations:

   **(a)** VLMLight relies solely on **visual perception**, whereas **ViRL-TSC performs multi-source information fusion**. Current VLMs still struggle to reliably extract fine-grained traffic attributes (e.g., exact vehicle counts) from images. To address this limitation, we design the **FM-VRE Module** to parse and construct multi-modal representations beyond raw visual signals.

   **(b)** The **decision logic** of the two frameworks is fundamentally different.
   – In VLMLight, the **VLM agent decides whether to call RL**.
   – In ViRL-TSC, the process is **RL → VLM**, meaning the RL agent produces an action first, and the VLM intervenes only when necessary.
   This design is crucial in real-world deployments: when communication failures occur, the **FM-VRE** and **FM-DEE** modules may not return results, yet the **RL agent has already produced an actionable decision** that can be executed immediately.

   Reviewers **6Xah** and **Za98** also suggested adding new scenarios and baselines. We incorporated **VLMLight** as an additional baseline and introduced a new **traffic-control intervention scenario**. Experimental results demonstrate that **combining VLMs with RL significantly improves robustness**, and **ViRL-TSC achieves consistently stronger performance**.

2. Regarding generalization to **multi-intersection** settings:
   Although coordinated control across intersections can indeed improve global efficiency, **single-intersection control remains the fundamental building block** of TSC research, as documented in prior work. It also provides a clean and controlled environment for analyzing model behavior—especially for our new RL–VLM integration mechanism.

   Nonetheless, we acknowledge the importance of scaling up. In future work, we plan to extend our **3D simulation environment** to multi-intersection networks. This presents considerable challenges for both algorithms and system design, as it requires integrating **SUMO multi-node networks** with fully synchronized **3D scene generation** to emulate real-world conditions.

[1] VLMLight: Traffic Signal Control via Vision-Language Meta-Control and Dual-Branch Reasoning. 2025 NeuIPS

---

### Author Response · Authors · 2025-12-04
**Review and Reviewer-Author Discussion Summary**

**Dear PCs, SACs, ACs, and Reviewers:**

We sincerely appreciate the reviewers’ time and constructive feedback. Although the initial scores (4/4/4/4) were slightly below the acceptance threshold, we have carefully revised the paper to address all raised concerns and hope that the updated version now meets the standard for acceptance. A summary of our revisions is provided below.

**Strengths**

All four reviewers agreed that integrating reinforcement learning (RL) with vision–language models (VLMs) for traffic signal control (TSC) represents a **meaningful, timely, and promising** research direction. This approach not only combines the efficient policy optimization capabilities of RL but also leverages the generalization ability of foundation models to enhance robustness in long-tail or previously unseen scenarios. Reviewers 6Xah and anK7 highlighted that incorporating VLMs into RL enhances robustness under long-tail or unseen scenarios, while Reviewer BKxt emphasized the ability of multimodal structures to handle complex real-world traffic scenes.

**Main Weaknesses**

In the initial review round, reviewers raised several concerns regarding the completeness of the evaluation and the clarity of the contribution. Specifically, Reviewer 6Xah noted that the experiments originally covered only a single long-tail scenario, while Reviewers Za98, BKxt, and anK7 pointed out the absence of comparisons with recent state-of-the-art methods. These limitations raised concerns about the sufficiency of empirical evidence and the novelty of the proposed method, leading all reviewers to assign an initial score of 4.

**Our Response and Improvements**

We sincerely appreciate the reviewers’ constructive suggestions. In response, we clarified the novelty of this work as the first TSC framework that combines RL with a VLM to process integrated **visual and multi-sensor** traffic data for decision-making. We expanded the experiments by adding a new **road-block control scenario**, and we further improved the comparison by including VLMLight (NeurIPS 2025) as an **additional baseline**. These updates provide a more comprehensive and current evaluation, better demonstrating the potential and performance of ViRL-TSC.


We addressed the reviewers’ comments through revisions and detailed responses. Unfortunately, due to special circumstances this year, we were unable to engage in deeper dialogue with the reviewers—something we genuinely regret, as their feedback was insightful and valuable. While future extensions such as multi-intersection coordination, cross-layout transferability, and large-model fine-tuning remain important research avenues, we believe the present work already offers substantial research contributions and meaningful practical value.


Thank you for your time and consideration.

**Authors**

---

### Meta-Review · Area_Chair_xuYx · 2026-01-08

**Summary:**

This paper proposes ViRL-TSC, integrating reinforcement learning with vision-language models for traffic signal control to improve robustness in long-tail scenarios. Reviewers agreed the direction is timely and promising, and the RL→VLM intervention design is reasonable. However, the submission received uniformly borderline scores (all 4s) due to concerns about limited experimental scope, single-intersection focus, high inference latency, and insufficient comparison with recent state-of-the-art methods. Questions were also raised about whether the demonstrated long-tail scenarios sufficiently represent real-world variability.

**Reviewer Concerns:**

The rebuttal added new scenarios and a recent baseline (VLMLight) and clarified design choices. While these improvements strengthen the paper, core limitations remain: experiments are still limited in scale, decision latency is high for real-time control, and generalization to coordinated multi-intersection settings is not demonstrated. Reviewers acknowledged the improvements but did not indicate score increases, and the work remains exploratory rather than conclusive.

**Reviewer Scores:**

All reviewers (4): Likely remain 4, as no reviewer signaled a clear upgrade after revisions.

---

### Decision · Program_Chairs · 2026-01-26

Reject